# The New Technologies Developed from Laser Shock Processing

**DOI:** 10.3390/ma13061453

**Published:** 2020-03-23

**Authors:** Jiajun Wu, Jibin Zhao, Hongchao Qiao, Xianliang Hu, Yuqi Yang

**Affiliations:** 1Shenyang Institute of Automation, Chinese Academy of Sciences, Shenyang 110016, Liaoning, China; wujiajun@sia.cn (J.W.); hcqiao@sia.cn (H.Q.); huxianliang@sia.cn (X.H.); yangyuqi@sia.cn (Y.Y.); 2Institutes for Robotics and Intelligent Manufacturing, Chinese Academy of Sciences, Shenyang 110169, Liaoning, China; 3College for Robotics and Intelligent Manufacturing, University of Chinese Academy of Sciences, Beijing 100049, China

**Keywords:** laser shock processing, stress effect, laser shock forming, warm laser shock processing, laser shock marking, laser shock imprinting

## Abstract

Laser shock processing (LSP) is an advanced material surface hardening technology that can significantly improve mechanical properties and extend service life by using the stress effect generated by laser-induced plasma shock waves, which has been increasingly applied in the processing fields of metallic materials and alloys. With the rapidly development of modern industry, many new technologies developed from LSP have emerged, which broadens the application of LSP and enriches its technical theory. In this work, the technical theory of LSP was summarized, which consists of the fundamental principle of LSP and the laser-induced plasma shock wave. The new technologies, developed from LSP, are introduced in detail from the aspect of laser shock forming (LSF), warm laser shock processing (WLSP), laser shock marking (LSM) and laser shock imprinting (LSI). The common feature of LSP and these new technologies developed from LSP is the utilization of the laser-generated stress effects rather than the laser thermal effect. LSF is utilized to modify the curvature of metal sheet through the laser-induced high dynamic loading. The material strength and the stability of residual stress and micro-structures by WLSP treatment are higher than that by LSP treatment, due to WLSP combining the advantages of LSP, dynamic strain aging (DSA) and dynamic precipitation (DP). LSM is an effective method to obtain the visualized marks on the surface of metallic materials or alloys, and its critical aspect is the preparation of the absorbing layer with a designed shape and suitable thickness. At the high strain rates induced by LSP, LSI has the ability to complete the direct imprinting over the large-scale ultrasmooth complex 3D nanostructures arrays on the surface of crystalline metals. This work has important reference value and guiding significance for researchers to further understand the LSP theory and the new technologies developed from LSP.

## 1. Introduction

Laser is one of the great inventions of natural science in the twentieth century. Since the successful production of the first laser equipment in 1960, the laser quickly received high attention in various fields and has developed rapidly, due to having valuable and special properties. Nowadays, laser is applied in almost all fields, such as industry, agriculture, medicine, national defense, and daily life, [1] etc. While the laser is applied in various field, a series of new technologies and cross-disciplines have been forming, such as nonlinear optics [2], optical communication technology [3], medical laser [4], tissue optics [5] and laser processing [6], etc. Laser processing is an interdisciplinary and comprehensive technology using the interaction of laser beams with matter to achieve the removal, connection and surface treatment of materials (including metallic materials and non-metallic materials). It can solve many industrial problems that cannot be solved by conventional processing methods, and at the same time, it can improve working efficiency and material processing quality [1,7,8,9].

With the rapid development of modern scientific technology and the high-end equipment manufacturing industry, the requirements for the mechanical properties of materials are becoming higher and higher, and the service environment of these materials are becoming harsher and harsher [1]. In order to improve the structural reliability and extend the fatigue life of materials without changing the matrix materials, the surface treatment technology plays an important role in industrial production. The common surface treatment technology mainly consists of surface chemical heat treatment, surface hardening, surface alloying, surface coating and shot peening, etc. In recent years, the surface treatment technology has been developed rapidly and is widely used in defense and civilian industry due to its significant effect without changing the original design [10]. Since the 1970s, lasers have been gradually applied to the surface treatment of metallic materials or alloys, which lead to many new surface treatment technologies, such as the laser surface hardening [11], laser chemical vapor deposition [12], laser physical vapor deposition [13], laser surface alloying [14], laser cladding [15] and laser shock processing (LSP) [16], etc. Compared with other laser treatment technologies, LSP utilizes the stress effect generated by the laser-induced shock waves rather than the laser thermal effect, which can create severe plastic deformation inside to the near-surface of metallic materials or alloys. As a result, the fatigue properties and service life of metallic materials or alloys are improved greatly [17]. For example, after the LSP treatment, the fatigue life of the vane-integrated disk can be improved about 4–6 times, which shows the great development prospect for LSP [18]. As a novel surface hardening treatment technology, LSP has the advantages of great hardening effect, high processing precision and good controllability of programming [19]. Compared with other normal surface hardening treatment technologies such as shot peening (SP), rolling and low plasticity burnishing, the depth of plastic deformation by LSP treatment can be reached to over 1mm, which is much higher than that by other methods with the depth of 75–250 μm at most [18,20]. In addition, LSP has the obvious advantages in maintaining the smooth surface morphology. For example, the 7075 Al alloy were treated by SP and LSP, the surface roughness of the SP-treated sample is 5.7 μm, while that of the LSP-treated sample is 1.1 μm [21]. Over decades of development, LSP has played an important part in the fields of aerospace manufacturing industry such as the extension of the fatigue life of vane-integrated disk and engine blades [22,23].

With the further development of laser equipment in recent years, LSP has been developed rapidly, and its application in material processing has been expanded continuously, too. As a result, many new technologies developed from LSP are appeared. In this work, the technical theory of LSP was summarized, which consists of the fundamental principle of LSP and the laser-induced plasma shock wave. The new technologies developed from LSP were introduced in detail from the aspect of laser shock forming (LSF), warm laser shock processing (WLSP), laser shock marking (LSM) and laser shock imprinting (LSI). This work has important reference value and guiding significance for researchers to further understand the LSP theory and the new technologies developed from LSP.

## 2. The Technical Theory of Laser Shock Processing

### 2.1. Fundamental Principle of LSP

Laser shock processing, also named as laser shock peening, is a novel material surface treatment technology by using the stress effect generated by laser-induced shock waves to improve the material properties, such as anti-fatigue, wear resistance and anti-stress corrosion, etc. [24,25]. The schematic of the principle of LSP is shown in Figure 1 [25]. Before the process of LSP, the LSP surface needs to be coated with an absorbing protective layer such as black tape, black paint or aluminum foil, and then covered with a transparent constraint layer such as water or optical glass, with the main purpose of improving the absorption rate of the laser energy for the metallic materials, and protecting the surface of the metallic materials or alloys from the laser thermal ablation. The laser beam with high power density and short pulse width passes through the transparent constraint layer and acts on the surface of metallic material, the absorbing protective layer that is coated on the surface of metallic material absorbs the laser energy, causing the material temperature to rise. The phenomenon of gasification and ionization will be occurred, and the expanding plasma with high temperature (>10^4^ K) and high pressure (>1GPa) is formed at about the same time. The plasma continues to absorb the laser pulse energy, and due to the restriction of the absorbing protective layer and constraint layer, the plasma expands rapidly and then explodes to form a high-pressure (GPa level) laser-induced plasma shock wave that acts on the surface of metallic material and propagates inside [21,24,25,26]. When the laser-induced plasma shock wave is loaded on the surface of metallic material, uniaxial stress will be generated along the propagating direction of the shock wave, which will lead to plastic deformation in the LSP area. After the laser-induced plasma shock wave is over (about tens of nanoseconds), the plastic deformation region will be limited and restricted by the surrounding material, so a biaxial compressive residual stress field will be generated, which is paralleled to the LSP surface [27,28]. The compressive residual stresses can eliminate the harmful tensile stresses due to the process of material processing such as mechanical machining, heat treatment, welding, laser cutting, plating and hard coating, etc. In addition, the compressive residual stresses can inhibit the initiation and expansion of fatigue cracks. Consequently, after the treatment of LSP, the mechanical properties of the metallic materials or alloys can be improved evidently [28]. The schematic diagram of compressive residual stress induced by LSP is shown in Figure 2 [28].

At present, the process structure of the constraint layer and absorbing protective layer is the most typical structure for the LSP treatment [29]. Whether being the constraint layer or the absorbing protective layer, its thickness should be suitable. The thickness of the constraint layer can affect the transmittance of the laser and the pressure of laser-induced shock waves, the lower thickness can increase the transmittance but easily cause the breakdown of the pressure, and the suitable thickness of the constraint layer such as water for most of the LSP process is about 1.2–2 mm [10,30]. The absorbing protective layer can improve the absorption rate of laser energy and increase the peak pressure of laser-induced plasma shock waves as well as protect the materials from thermal melting. The related research shows that there exists a critical thickness of the absorbing protective layer, and that higher thickness can lead to the pressure losses of laser-induced plasma shock waves, but lower thickness can lead to the thermal melting in the near-surface layer of material and the formation of rougher impact pits, and the common thickness of the absorbing protective layer such as black tape is about 100 μm [1,31]. Therefore, in the LSP treatment, the suitable thickness of the constraining layer and absorbing protective layer can improve the transmittance and pressure of the laser-induced plasma, resulting in obtaining a better hardening effect [32].

The common feature of surface treatment technologies is to utilize the severe plastic deformation in the near-surface of metallic materials or alloys to get the result of material surface strengthening [33]. In contrast to the traditional surface treatment technologies, the LSP utilizes the laser-induced plasma shock wave, which increases the pressure of the shock wave by the typical structure of the absorption protection layer and constraint layer in general. Compared to other surface treatment technology, LSP has the advantages of ultra-high pressure, ultra-high shock wave mechanical energy and ultra-high plastic strain rate, so the strengthening effect is better. In addition, the spot size and position of the pulsed laser can be adjusted precisely by the industrial robot, so LSP is suitable to strengthen the structures with complex shapes or shapes that are difficult to handle with traditional processes [27].

### 2.2. The Laser-Induced Plasma Shock Wave

The principle of LSP tell us that the propagation of laser-induced plasma shock wave in the metallic materials or alloys can cause high strain rate dynamic responses in the near-surface of material, thereby realizing the surface strengthening of metallic materials or alloys [10]. Therefore, the laser-induced plasma shock wave is the direct power for material hardening. So, herein is the investigation for the forming process and calculating model of the laser-induced plasma shock wave, which will provide the theoretical basic for researchers to explore the technological theory of LSP deeply.

In the irradiation of a strong pulsed laser, which will lead to the partial heating and ionization of metallic materials or alloys, then the further heating and ionization of metallic materials or alloys will occur by means of the transportation mechanism such as heat conduction and thermal radiation. At this time, the laser-supported combustion waves (LSCW) are formed. Subsequently, partial lasers still pass through the plasma volume and act on the surface of metallic materials or alloys, the radiation of plasma that is near the LSP region will be helpful to enhance the thermal coupling of laser and metallic target, that will lead to the formation of laser-supported detonation waves (LSDW), which can absorb the laser pulse energy completely [10,34,35,36,37]. The structure diagram of LSCW and LSDW is shown in Figure 3 [1].

The formation process of LSDW is very complicated, which is closely related to the characteristics of pulsed laser, metal target, constrained layer, and absorption layer, etc. The aerodynamic model of LSDW was first proposed by Paйɜep in 1965 [1]. In his model, the LSDW can be considered as a strong fault surface wave (namely shock wave) without thickness and which propagates at supersonic speed, and the incident laser would be absorbed completely and converted into a power that propels the wave forward. The hydrodynamic equations of LSDW can expressed as the following equations [1,10].
(1)$$\nu_{2}=\frac{\nu_{1}}{\gamma_{b}+1}$$
(2)$$\rho_{2}=\frac{\gamma_{b}+1}{\gamma_{b}}\rho_{1}$$
(3)$$E_{u2}=\frac{\gamma_{b}{\left[2\left(\gamma_{b}{}^{2}-1\right)\right]}^{\frac{2}{3}}}{\left(\gamma_{b}+1\right)\left(\gamma_{b}{}^{2}-1\right)}\rho_{1}{}^{\frac{2}{3}}I^{\frac{2}{3}}$$
(4)$$p_{2}=\frac{{\left[2\left(\gamma_{b}{}^{2}-1\right)\right]}^{\frac{2}{3}}}{\gamma_{b}+1}\rho_{1}{}^{\frac{1}{3}}I^{\frac{2}{3}}$$
where, *p*_1_, *ρ*_1_ and *E_u_*_1_ are the pressure, mass density and internal energy of wave front gas of LSDW, respectively, and the *p*_2_, *ρ*_2_ and *E_u_*_2_ are the pressure, mass density and internal energy of wave back gas of LSDW, respectively. *v*_1_ is the velocity of LSDW, *v*_2_ is the particle velocity of wave back gas of LSDW, *I* is the laser power irradiance, and *γ*_b_ is the adiabatic exponent of plasma [1].

According to the Equations (1)–(4), we can know that the pressure of LSDW is proportional to the laser power irradiance.

The acting time of laser-induced plasma shock wave on the metallic material is extremely short, which is within tens of nanoseconds in general. In order to calculate the pressure of laser-induced plasma shock wave during LSP, a number of calculation model of shock wave pressure were proposed. Among them, one-dimensional model of shock wave pressure proposed by Fabbro [38] is well known and most universally acknowledged, which can reflect the change law of the peak pressure of laser-induced plasma shock waves with the laser pulse energy. The calculation formula for the peak pressure of laser-induced plasma shock waves is as follows:(5)$$P_{\max}\left(GPa\right)=0.01{\left(\frac{\xi }{2\xi +3}\right)}^{0.5}z^{0.5}\left(g\cdot cm^{-2}\cdot s^{-1}\right)I^{0.5}\left(GW\cdot cm^{-2}\right)$$
where, *P*_max_ is the peak pressure of laser-induced plasma shock wave; *ξ* is the fraction of absorbed laser pulse energy (typically equals to 0.1), and *I* is the laser power irradiance. Z is the combined acoustic impedance of the metallic material and the confinement layer, which can be calculated by the following relation: (6)$$\frac{2}{Z}=\frac{1}{Z_{1}}+\frac{1}{Z_{2}}$$
and the acoustic impedance of the metallic material and the constraint layer can be expressed as [39]:(7)$$Z_{i}=\rho_{i}D_{i}$$
where I = 1 or 2, which represent the metallic material and constraint layer, respectively. *ρ*_i_, *D*_i_ are the mass density and the propagation velocity of laser-induced plasma shock waves, respectively.

The calculation formula of laser power irradiance can be expressed as follows [40]:(8)$$I=\frac{4E}{\pi D^{2}\tau }$$
where, *I* is the laser power irradiance, *E* is the laser pulse energy, *D* is the laser spot diameter, and *τ* is the laser pulse width.

In the one-dimensional model of plasma shock wave, we can consider that the plasma shock wave propagates into an elastic perfectly plastic metal half space. This model considers that the plastic deformation and residual stress can be occurred only when the peak pressure of laser-induced plasma shock wave exceeds the Hugoniot elastic limit (*HEL*). The *HEL* can be given by the following equation.
(9)$$HEL=\left(1+\frac{\lambda }{2\mu }\right)Y$$
where *Y* is the dynamic yield strength, *λ* and *μ* are the Lame constants, which can be expressed as the following equations: (10)$$\lambda =\frac{E\nu }{\left(1+\nu \right)\left(1-2\nu \right)}$$
(11)$$\mu =G=\frac{E}{2\left(1+\nu \right)}$$
where *ν*, *E* and *G* are the Poisson’s ratio, elastic modulus and shear modulus, respectively. In general, the dynamic yield strength is about 2–4 times of the static yield strength [41].

We have investigated the spatial distribution of the laser-induced plasma shock wave, which is meaningful for researchers studying the residual stress distribution and hardness distribution of metallic materials or alloys after the LSP treatment. For the laser beams with single transverse mode or the single longitudinal mode, the laser energy or the laser power irradiance obeys the Gaussian spatial distribution, which can be expressed as following equation [1,41]: (12)$$I\left(r\right)=I\cdot e^{-\frac{2r^{2}}{R}}$$
where *R* is the radius of laser spot, which is defined as the size at which the light intensity drops to the 1/e^2^ of maximum. And *r* is the radial distance from the center of the laser beam.

According to the work of Zhang [42], the pressure of laser-induced plasma shock wave also obeys the Gaussian spatial distribution, but with its 1/e^2^ radius equal to $\sqrt{2}R$. The pressure of shock wave in temporal and spatial distribution can be expressed as follows: (13)$$P\left(r,t\right)=P\left(t\right)exp\left(-\frac{r^{2}}{2R^{2}}\right)$$
where *P*(t) is the spatially uniform pressure of laser-induced shock wave that changes with time.

According to the Equation (13), we can obtain the spatial distributions of laser power irradiance and peak pressure of the shock wave (as shown in Figure 4) [1]. Compared with the spatial distribution of laser power irradiance, the distribution of peak pressure of the laser-induced plasma shock wave is more uniform in two-dimensional space due to the increase of its 1/e^2^ radius [42].

As observed from Equation (5), we can know that the peak pressure of shock wave is proportional to the square root of laser power irradiance. In addition, the laser power irradiance obeys the quasi-Gaussian distribution over time, so the peak pressure of shock waves also obey the quasi-Gaussian distribution over time [24]. The related reference [43] tells us that the pulse width of the shock waves load is approximately 2–3 times that of laser pulse width.

## 3. The New Technologies Developed from LSP

LSP is a novel material surface modification technology that can significantly improve the mechanical properties and extend service life by using the stress effect generated by laser-induced plasma shock waves. With the development of intelligent manufacturing in recent years, many new technologies developed from LSP have emerged, which broadens the application of LSP and enriches its technical theory. In this section, we importantly introduce the new technologies developed from LSP from the aspect of laser shock forming, warm laser shock processing, laser shock marking and laser shock imprinting.

### 3.1. Laser Shock Forming

Laser shock forming (LSF) is a complex dynamic forming process, which is developed from LSP. As a purely mechanical forming method, LSF is utilized to modify the curvature of metal sheet by thermal stress that induced by the irradiation of laser beam [44]. The schematic of the LSF process is shown in Figure 5. The surface of metal sheet needs to be coated with an absorption protection layer firstly, and then covered by the flowing water as the constraint layer. So, a typical structure of absorption protection layer and constraint layer is formed, which is the same as for the LSP. The pulsed laser with high-intensity passes through the constraint layer and ablates the absorbent layer and scans along a designed path. The generated laser-induced plasma shock wave applied to the metal sheet, which will lead to local plastic deformation and material expansion in the shocked area, causes the plastic bending of the metal sheet. A number of incremental deformations induced by LSP applied successively to the metal sheet are accumulated to obtain greater convex or concave bending deformation, which is depending on the process parameters such as the laser pulse energy and the thickness of metal sheet [45,46]. It should be noted that during LSF, the bottom is not restricted, just one end needs to be clamped, which is different from the LSP. In order to ensure the quality of LSF, the pulsed laser beam needs to be perpendicular to the surface of the metal plate during LSF, so the serpentine shocked path and the scanning direction from free end to clamping end are selected generally (as shown in Figure 5b).

Studies about LPF can be traced back to 2002; the three-dimensional bending principle and precise forming method of the metal plate by using the LSP system was first proposed by the Lawrence Livermore National Laboratory (USA) [47]. With the deepening research on LSF, the bending deformation for different materials are discussed and the corresponding deformation mechanisms are proposed. In 2010, the stress gradient mechanism (SGM) and shock bending mechanism (SBM) of LSF (as shown in Figure 6 and Figure 7, respectively) were proposed by Hu et al. [46]. When the metal sheet is thick or the laser power irradiance is low, the laser-induced plasma shock wave cannot penetrate the metal sheet, and this situation is the SGM, which makes the LSP region compressed at the thickness direction and puts the material beneath the top surface into the tensional state at the length direction, and causes a negative bending moment M that drives the metal sheet downward. As a result, the downward bending of the convex is generated at the free end of the metal sheet [46,47]. When the metal sheet is thin enough or the laser power irradiance is high enough, the laser-induced plasma shock wave can penetrate the metal sheet without significant attenuation, and this situation is the SBM, which makes the LSP region plastically deformed completely due to the pressure of laser-induced plasma shock wave being very high, and it can be considered that there is almost no steep stress gradient through the thickness of the metal sheet. So, the downward movement will exist through the thickness at the LSP region of metal sheet and cause a positive bending moment *M* that drives the metal sheet upward. As a result, the upward bending of concave is generated at the free end of metal sheet [46,47].

According to the mechanism of LSF, we can know that, if the metal sheet is thick enough or the laser power irradiance is low enough, the metal sheet is hardly bent, so the compressive residual stress will be formed in the near-surface of metal sheet, which is the same as LSP. In contrast, if the metal sheet is thin enough or the laser power irradiance is high enough, it will be transformed into laser deep drawing (as shown in Figure 8) [46]. So, it can realize the bending deformation from one direction to another direction smoothly by setting different process parameters. LSF is a novel forming technology, through the laser-induced high dynamic loading which can modify the curvature of metal sheet with the advantages of being tool-free, having high efficiency and with non-contact, [48,49] etc. At the same time, as a pure cold-work process, LSF can generate compressive residual stress over both side of metals, which can resist cracks from corrosion and fatigue for formed parts.

### 3.2. Warm Laser Shock Processing

LSP has been widely applied to the industrial fields due to LSP having the advantages of the introduction of compressive residual stress layer with a high values and deep depth, the adaptability to treat complex parts and the controllability of laser parameters (such as the laser pulse energy, laser spot size, laser shape, laser pulse width and working frequency of laser equipment) [50]. However, under the mechanical loading or the thermal heating process, the compressive residual stress and the work-hardened layer will be relaxed, which will restrict the application of LSP [50,51,52]. According to the research of Ren [53] on the Ni-based alloy GH4169 after the treatment of LSP, its compressive residual stress induced by LSP was relaxed at high temperature, and the degree of compressive residual stress thermal relaxation was increased with the temperature, and this similar phenomenon can also be found in the study of Zhou [52].

In order to tackle the challenge of LSP, warm laser shock processing (WLSP) is developed by Ye and Liao [54]. WLSP is an innovative thermal mechanical surface hardening technology developed from LSP, which combines the advantages of LSP, dynamic strain aging (DSA) [55] and dynamic precipitation (DP) [56] to obtain the distinctive high stable micro-structures. The DSA take advantages of the hardening effect of high density dislocation, dislocation locking, grain refining and so on by the interaction between the moving solute atoms of metallic materials or alloys and the moving dislocation, which can improve the cyclic stability and thermal stability of micro-structures of metallic materials or alloys significantly, and reduce the compressive stress relaxation at high temperature or under the cyclic loading [55,57]. DP is the generation of precipitations during the plastic deformation process, which will lead to the nucleation of precipitates and promote material hardening. As a result, the mechanical performance and fatigue life of metallic materials and alloys are further improved by the treatment of WLSP.

The schematic diagram of WLSP is shown in Figure 9 [50,58]. Prior to the treatment of WLSP, the shocked surface of metallic materials or alloys should be coated with a heat-resistant absorbing layer (such as aluminum foil) with a uniform thickness, and the transparent silicone oil is used as the confining layer usually, which are similar to the typical structure of LSP. The shocked target material should be heated up to a certain temperature by a heating system below the sample fixture. The laser-induced plasma shock wave with high pressure and high strain rate acts on the shocked surface of heated sample, the thermal-mechanical coupling effect will occur, which can not only promote the formation of a more stable dislocation structure than the treatment of LSP but also lead to the nano-precipitation at the grain boundaries. So, the strengthening effect by WLSP is better than that by LSP [59].

The enhanced mechanical properties such as the hardness, residual stress and the stability of compressive residual stress are the key beneficial surface properties induced by the treatment of WLSP [50] according to the WLSP experiments of aluminum alloy 6061 by Liao [60] and AISI 1042 steel by Tani [61] (as shown in Figure 10a,b, respectively). From the Figure 10, we can know that with the treatment of LSP and WLSP, the surface hardness of the samples are enhanced. However, compared with the samples treated by LSP, the samples treated by WLSP own a higher surface hardness. For instance, with the laser parameters of 1.6 GW/cm^2^ for the laser power irradiance, 5 ns for the laser pulse width and 2 mm for the diameter of laser spot (the main experiment parameters are listed in Table 1, the same below), the surface hardness of aluminum alloy 6061 treated by WLSP at the temperature of 160 °C (132VHN) is about 44% higher than that treated by LSP (94VHN). In addition, the surface hardness of aluminum alloy 6061 shocked by WLSP at the temperature of 160 °C is higher than that shocked by WLSP at the temperature of 90 °C. With the laser parameters of 1.5 GW/cm^2^ for the laser power irradiance, 8 ns for the laser pulse width and 4 mm for the diameter of the laser spot, the surface hardness of AISI 1042 steel shocked by WLSP at the temperature of 500K (321HV) is about 8% higher than that shocked by LSP (298HV). The enhanced surface hardness is attributed to the strain hardening effect by the severe plastic deformation of near surface material and the precipitation hardening effect by the generation of high density nano-precipitates [50].

Similar to the surface hardness, the residual stress of samples treated by WLSP are higher than that treated by LSP, according to the studies of Ye (as shown in Figure 11 for detail) [62,63]. From the Figure 11, we can know that for AA7075 or AA6160, the surface compressive residual stress and the residual stress layer depth of samples treated by WLSP are higher than that treated by LSP. Besides the residual stress, the stability of compressive residual stress is also improved, which is of great significance for the extended fatigue life of materials. Figure 12 shows the effects of WLSP on the low-cycle fatigue behavior of Ti6Al4V alloy [64]. As observed in the Figure 12, the samples treated by WLSP show a higher stability of residual stress and the axial displacement than that treated by LSP. However, it doesn’t mean that the higher the temperature of WLSP, the better the stability of residual stress and the axial displacement. For instance, the residual stress relaxation in the first 10 cycles (as shown in Figure 12a), and the residual stress at 10 cycles of samples treated by LSP or WLSP at the temperatures of 20, 100, 200, 300 and 400 °C are 16, 42, −19, −86 and −49 MPa, respectively. This similar phenomenon can be also found by the studies of Ye [62] and Huang [65]. So, there exist the most suitable WLSP temperatures to obtain the best stability of residual stress and the highest compressive residual stress. According to the DSA theory proposed by Qian (“Qian-Xiao-Li model”) [66], the suitable WLSP temperature should be 0.2–0.5 *T*_m_ (*T*_m_ is the melting temperature of metallic material or alloy). In addition, according to the research of Meng [67], the optimum pressure of laser-induced plasma shock wave for WLSP treatment is 2 times of *HEL*, which is different from the best pressure of laser-induced plasma shock wave for LSP treatment with 2.5 times of *HEL*.

The principle of LSP and WLSP tell us that the enhanced mechanical performance of the material is very closely related to the micro-structure. The transmission electron microscope (TEM) is usually used to investigate the micro-structures evolution. The TEM image for grain structure of AA6061-T6 near the top surface after LSP and WLSP is shown in Figure 13 [63]. Compared with LSP sample, the nanoscale precipitates appeared in the WLSP sample, which can be blamed by the effect of DSA. At the same time, there are no second phase particles that can be found in the LSP sample [50,63]. During WLSP, the effect of DSA can serve the thermal energy to promote the forming of nanoscale precipitates with the stress effect of the ultra-high strain rate that is generated by the laser-induced plasma shock waves, which can supply numerous nucleation situations for the precipitation [63]. Then these nucleated nanoscale precipitates will be interacted with dislocations, which will lead to the high-density dislocation arrangement and the nanoprecipitates. However, LSP does not have the effect of DSA, so the lower density precipitates occur in LSP treated samples, and the high density nano-precipitates occur in the WLSP treated samples, and the high density nano-precipitates can facilitate the pinning of dislocations. So, the WLSP-treated samples will have a greater stability of dislocation structure. The increase of surface hardness, compressive residual stress, depth of residual stress layer and the stability of dislocation structure, which will inhibit the generation and propagation of cracks, results in the improved the stability of residual stress [68]. Compared with the LSP-treated samples, the compressive residual stress lever of the WLSP-treated sample is higher than that, and the effect of DA and DSA can provide the key contribution to the improvement [63,69]. In conclusion, the material strength and stability of residual stress by WLSP treatment are higher than that by LSP treatment.

### 3.3. Laser Shock Marking

The stress effect generated by the laser-induced plasma shock wave is usually used to enhance the mechanical performance, precise shaping and enhance stability of mechanical performance of materials [70], such as for the LSP, LSF and WLSP. In the field of metal materials marking, the traditional methods are achieved by etching and engraving in general. Laser marking (LM) is a well-accepted method that can be applied to provide the visualized marks on the materials’ surface by utilizing the thermal effect induced by the laser with high energy to make the marking region melt locally and obtain the designed marking shape [71]. However, there exist some disadvantages in the LM, such as the oxidation of material and the presence of the remelted layer. In order to overcome the disadvantages of current marking technologies, Lu et al. [72] developed a laser shock marking (LSM) technology, which is developed from LSP. LSM utilize the stress effect generated by the laser-induced plasma shock wave rather than the laser thermal effect. As a novel marking method, LSM combines the advantages of LSP and LM.

The schematic diagram of LSM is shown in the Figure 14 [72]. Prior to the LSM, the marking surface of the material needed to be coated with an absorbing protective layer (such as black tape), and the thickness should be no less than the critical thickness of the LSP, as we know, with the stress effect generated by laser-induced plasma shock wave, which will lead to concave morphology on the surface of the material. The shape of the concave morphology is the same as the laser spot and its size is slightly larger than the diameter of the laser spot. In order to obtain the desired marking shape, before the LSP treatment, a second absorbing protective layer with a square hole is devised and applied on the first absorbing protective layer. It should be noted that the size of square hole needs to be smaller than the laser spot in size, and the laser beam needs to be aligned with the square hole area [72]. As a result, a square concave morphology that corresponds to the designed square hole can be obtained. According to the main functions of the absorbing protective layer, the first absorbing protective layer is named as the “shape forming layer”, and the second absorbing protective layer that reduces the pressure of the laser-induced plasma shock wave, is named as the “shape controlling layer”. As shown in the Figure 14, the thickness of the second absorbing protective layer is thicker than that of first absorbing protective layer. So, the pressure induced by the laser plasma shock wave inside the square hole is higher than outside the square hole, and the concave morphology inside the square hole is also higher than outside the square hole. Based on the principle, a series of concave morphology induced by LSP successively to the marking surface of metallic materials or alloys can be accumulated to obtain the designed marking shape by means of the multi-spot LSP treatment.

The “X” letter marked by the LSM technology is shown in Figure 15, which is the unique application of LSM currently [72]. Where the experimental samples were 7075 aluminum alloy with the size of no less than 24 × 24 × 4 mm, the “ shape forming layer ” was the black insulation tape with the thickness of 130 μm (one layer), and the “shape controlling layer” was also the black insulation tape, but its thickness was 390 μm (three layer). The overlapping rates laser spot in the marking area was 30%, and the laser pulse energy was 8.5 J, while the laser pulse width was 18 ns, and the diameter of the laser spot was 3.6 mm. The constraint layer in the experiment was flowing water with the thickness of 2 mm.

As observed from the Figure 15, the “X” letter mark can be seen clearly. In the “shape controlling” marking pattern (Figure 15a shows the 2D surface topography, Figure 15b shows the 3D surface topography), the X-shape is in the form of “skull”, which is the matrix region of material without LSP treatment actually, and its height is higher than that of the surrounding region of material. In the “shape forming” marking pattern (Figure 15d shows the 2D surface topography, Figure 15e shows the 3D surface topography), the X-shape is in the form “embossed” on the surface, and its height is lower than that of the surrounding matrix region of material.

The designed marking shape on the surface of metallic materials or alloys can be obtained by the LSM method. Compared with other marking technologies of metallic materials or alloys, the visual effects of samples by the LSM technology is relatively poor. In addition, due to the difficulty in the preparation of absorbing layer with a designed shape and suitable thickness, the accuracy of the marking shape by LSM in size will be relatively lower than that by other methods [22,72]. However, with the stress effect generated by the laser plasma shock wave, the marking region by LSM can maintain super mechanical properties and higher fatigue life [22].

### 3.4. Laser Shock Imprinting

In recent years, with the rapid development of the electronics, biosciences, medical device and micro-electro-mechanical system, the demand for the micro-nano products with the multifunctional features is increasing, and the processing of the microscale or nanoscale metallic structures related to that is also becoming more and more important, which promotes the development of metal micro-nano forming technology [73]. For some ultra-precise components, one of the challenging works is the large-scale manufacturing of a special 3D nanostructure with high fidelity and quality [74]. At present, both the nanoimprint lithography [75] and the direct nanoimprinting of metallic nanoparticles [76] have proved to be useful methods of manufacturing the 3D nanostructure on polymers and metallic glasses. However, the forming limitations such as size effects in plasticity, grain size effects and the fluctuations of plasticity at the nanoscale caused by localized dislocation bursts, etc., limits the feature size to be larger than the grains. In order to solve these limitations, methods that use a nanocrystal line, heating samples close to melting temperature, or that use a super-hard mold are selected in general [74]. However, these methods cannot solve the problems of the obtained products with serious drawbacks.

Laser shock imprinting (LSI) is a novel nanoshaping method that is developed from LSP, which was first completed by Cheng [74,76]. It utilizes the stress effect generated by laser-induced plasma shock wave to press the thin metallic layers into a silicon nanomold with a special 3D nanostructure. According to the related report [60], after using the nanomolds over 100 times, its performance has not damaged or degenerated. Due to the ultrahigh strain rate deformation induced by LSP, LSI has the ability to overcome the limitations for the nanoscale forming of coarse-grained metals, and realizes the large-scale nanoshaping of ultrasmooth 3D crystalline metallic structures [74,76].

The schematic illustration of LSI is shown in Figure 16 [74]. The Nd:YAG laser pulse with the laser power irradiance of 0.3–1.4 GW/cm^2^, wavelength of 1064 nm and laser pulse duration of 5 ns passes through the glass and irradiates the sample surface that is coated with graphite ablation layer (thickness of 10 mm). The interaction between the laser pulse and the graphite ablation layer will lead to plasma with high temperatures and high pressure. Due to the restriction of the confining media (glass), the plasma expands rapidly and then explodes, generating a strong laser-induced plasma shock wave. With the stress effect generated by laser-induced plasma shock wave, the metallic sheet is compressed into the underlying nanomold (Si mold) and attached to the substrate surface firmly [22,74,76].

Figure 17 shows several examples of complex 3D nanostructures manufactured with the LSI method. Figure 17a shows the array of nanogears imprinted on cold-rolled aluminum foil, and the size of gear tooth (with the length of 300 nm, height of 50 nm, width of 200 nm and radius at tooth bottom of 50 nm) are significantly smaller than the original grain size of aluminum foil (1.4 mm). Figure 17b shows the array of triangular V-grooves with aspect ratios of 1:5 (with the width of 100 nm, height of 500 nm and V-angle of 14°). Figure 17c shows the array of the ultrasmooth nanotrenches fabricated on the Ti surface layer with the thickness of 2.5 mm [74]. These observations can be concluded that LSI, at the high strain rates induced by LSP, has the ability to complete the direct imprinting over the large-scale ultrasmooth complex 3D nanostructures arrays on the surface of crystalline metals by the laser-generated stress effect [74]. In addition, LSI has the ability to overcome the limitations of nanoforming and the nanoscale size effect of metallic materials. As a high-throughput 3D nanoscale imprinting technique, LSI has the ability to utilize the laser pulse to manufacture the nanoscale metal crystal structures on a 6-inch surface within 30 s, and has potential in the development of future electronics, optics, and sensing devices [22,74].

## 4. Conclusions

LSP is an advanced material surface hardening technology using the laser-generated stress effect, which can significantly improve mechanical properties and extend service life. Based on the laser-generated stress effect, many new technologies developed from LSP have emerged. This paper has summarized the technical theory of LSP and introduced the new technologies developed from LSP from the aspect of LSF, WLSP, LSM and LSI, which has important reference value and guiding significance for researchers to further understand the LSP theory and the new technologies developed from LSP.
(1)During LSP, the laser-generated stress effect can lead to severe plastic deformation, which will create the introduction of compressive residual stresses inside the material and the evolution of micro-structures. As a result, with LSP treatment, the mechanical properties and fatigue life of metallic materials or alloys are improved significantly.(2)Similar to the fundamental principle of LSP, the new technologies developed from LSP also utilize the laser-generated stress effect. LSF is utilized to modify the curvature of the metal sheet through the laser-induced high dynamic loading, and the metal sheet is found to be a convex or concave deformation that is depending on the sheet thickness and the laser power irradiance. As a thermal-mechanical surface hardening technology, WLSP combines the advantages of LSP, DSA and DP, so the material strength and the stability of residual stress and micro-structures by WLSP treatment are higher than that by LSP treatment. LSM is an effective method to obtain the visualized marks on the surface of metallic materials or alloys, and its critical aspect is the preparation of the absorbing layer with a designed shape and suitable thickness. At the high strain rates induced by LSP, LSI has the ability to complete the direct imprinting over the large-scale ultrasmooth complex 3D nanostructures arrays on the surface of crystalline metals.


## Figures and Tables

**Figure 1 materials-13-01453-f001:**
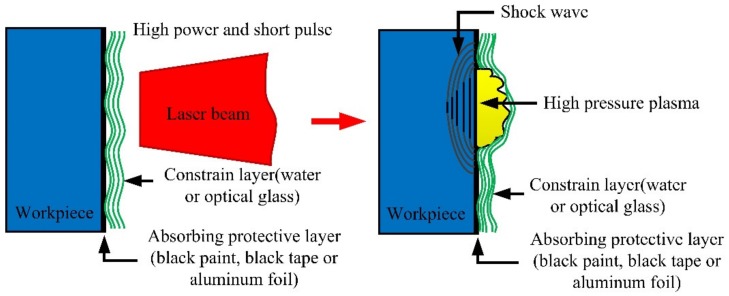
The schematic of the principle of laser shock processing (LSP) [25].

**Figure 2 materials-13-01453-f002:**
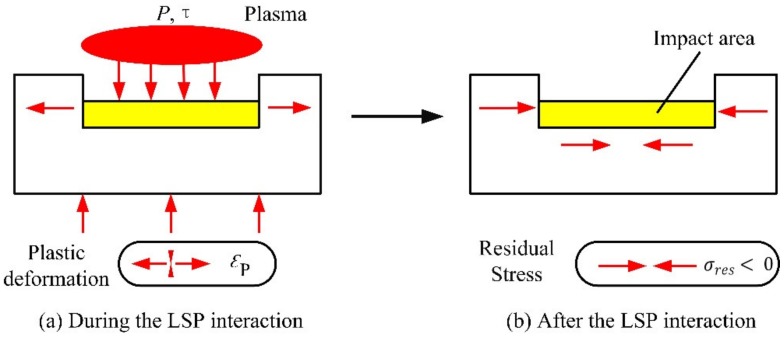
The schematic diagram of compressive residual stress induced by LSP [28].

**Figure 3 materials-13-01453-f003:**
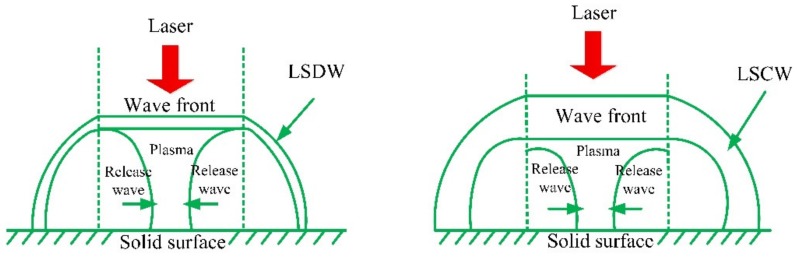
The structure diagram of LSCW and LSDW [1].

**Figure 4 materials-13-01453-f004:**
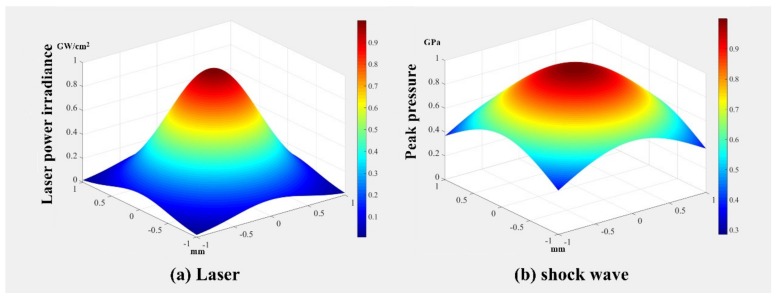
Spatial distributions of laser power irradiance and peak pressure of shock wave [1].

**Figure 5 materials-13-01453-f005:**
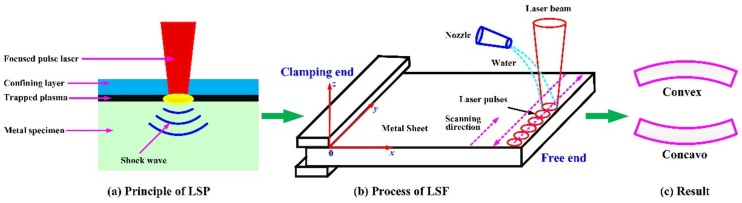
The schematic of LSF process.

**Figure 6 materials-13-01453-f006:**
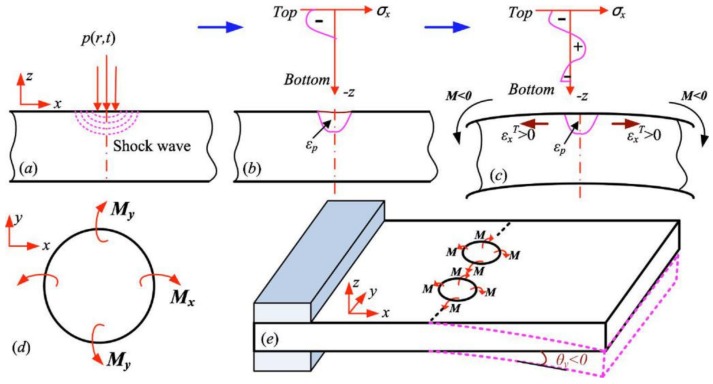
The stress gradient mechanism (SGM) of LSF: (**a**) LSP loading; (**b**) generation of stress gradient; (**c**) relaxation of stress gradient and bending; (**d**) bending moment on the cell and (**e**) bending of metal sheet with one end fixed [46].

**Figure 7 materials-13-01453-f007:**
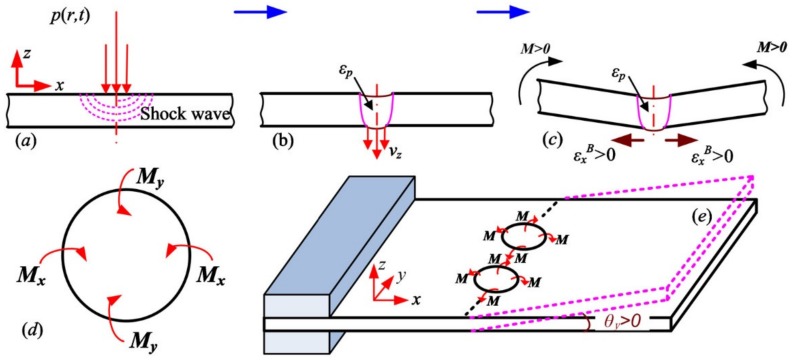
The shock bending mechanism (SBM) of LSF: (**a**) LSP loading; (**b**) downward movement and plastic deformation; (**c**) shocked bending; (**d**) bending moment on the cell and (**e**) bending of metal sheet with one end fixed [46].

**Figure 8 materials-13-01453-f008:**
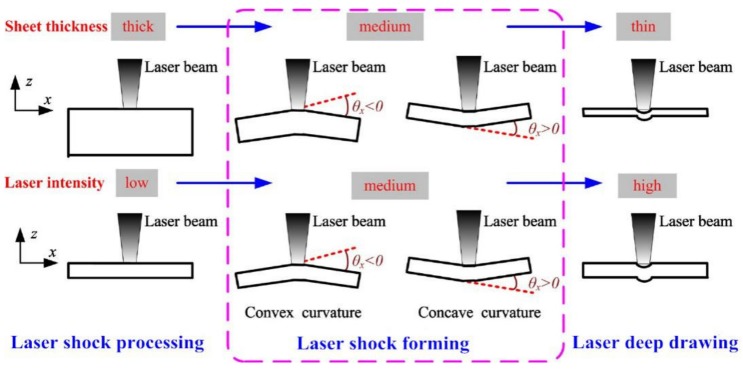
The effect of sheet thickness and laser intensity on the coupling of two mechanism of LPF [46].

**Figure 9 materials-13-01453-f009:**
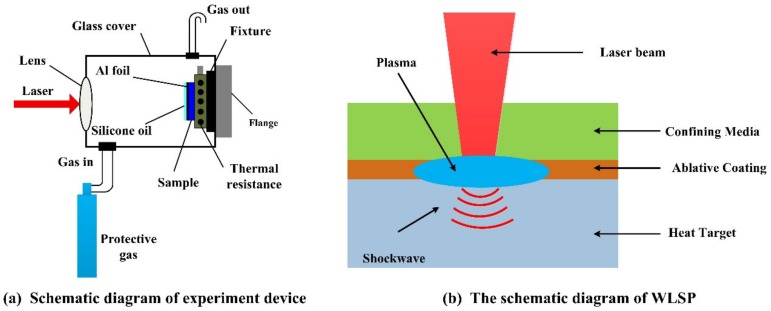
The schematic diagram of warm laser shock processing (WLSP) [50,58].

**Figure 10 materials-13-01453-f010:**
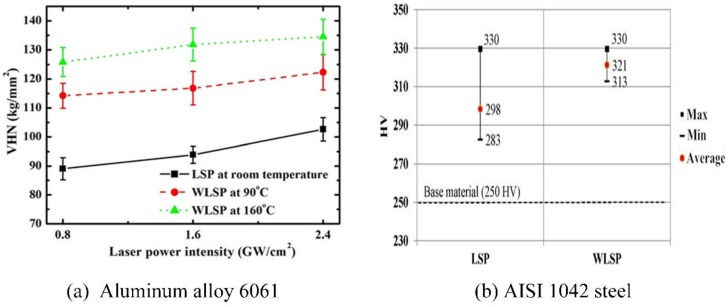
The enhanced surface hardness by WLSP: (**a**) aluminum alloy 6061, (**b**) AISI 1042 steel [60,61].

**Figure 11 materials-13-01453-f011:**
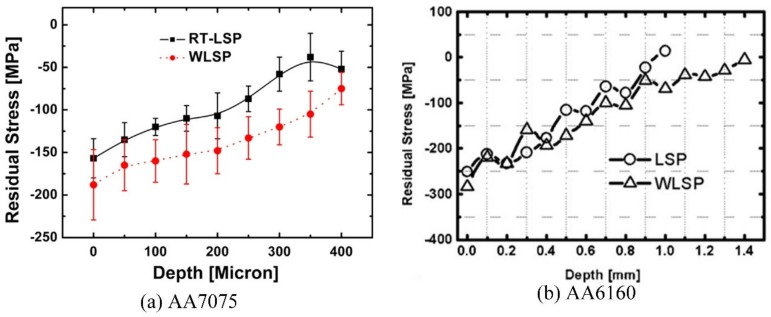
The enhanced residual stress by WLSP: (**a**) AA7075, (**b**) AA6160 [62,63].

**Figure 12 materials-13-01453-f012:**
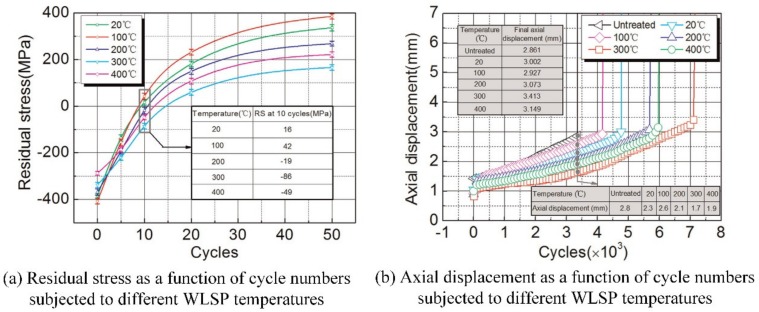
The effects of WLSP on the low-cycle fatigue behavior of Ti6Al4V alloy [64].

**Figure 13 materials-13-01453-f013:**
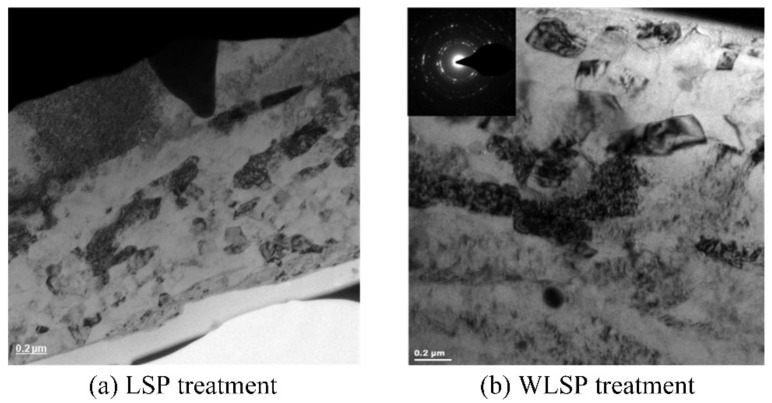
The TEM image for grain structure of AA6061-T6 near the top surface after LSP and WLSP [63].

**Figure 14 materials-13-01453-f014:**
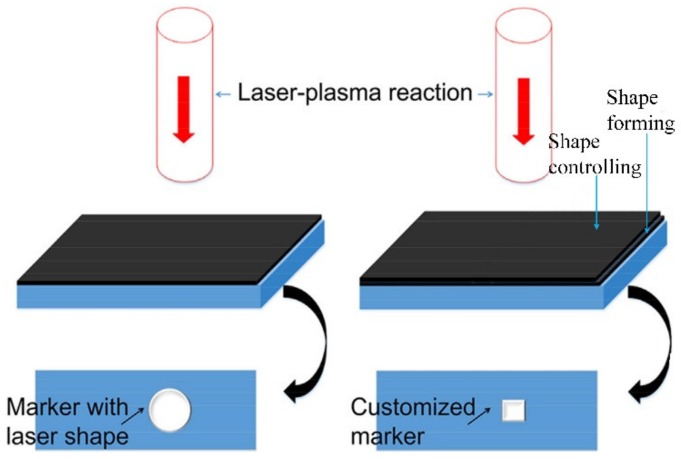
The schematic diagram of LSM [72].

**Figure 15 materials-13-01453-f015:**
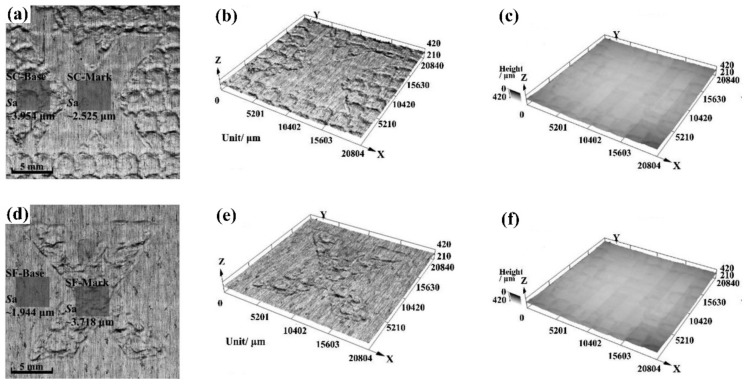
The “X” letter marked by the LSM technology. (**a**–**c**) “shape controlling” marking pattern; (**d**–**f**) “shape forming” marking pattern [72].

**Figure 16 materials-13-01453-f016:**
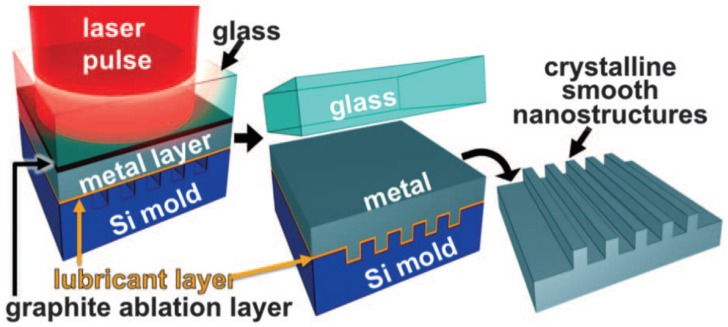
The schematic illustration of LSI [74].

**Figure 17 materials-13-01453-f017:**
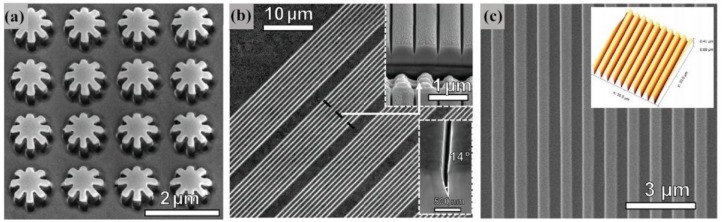
The SEM images of several complex 3D nanostructures manufactured with LSI. (**a**) Array of nanogears imprinted on cold-rolled aluminum foil; (**b**) array of triangular V-grooves; (**c**) array of the ultrasmooth nanotrenches fabricated on the Ti surface layer with the thickness of 2.5 mm. [74].

**Table 1 materials-13-01453-t001:** Main experiment parameters of LSP and WLSP provided by selected references reports in Section 3.2.

References	Material	Type of Laser System	Laser Parameters						Absorbing Layer	Confining Layer	Temperature	Materials Parameters
			Wavelength (nm)	Pulse Width (ns)	Power Irradiance (GW/cm^2^)	Diameter (mm)	Repetition Rate (Hz)	Overlap Rate				
Liao [60]—Figure 10a	AA6061	ND-YAG	1064	5	0.8–2.4	2	-	75%	Al foil	BK7 glass	20–160 °C	Hardness
Tani [61]—Figure 10b	AISI 1042	Nd:YAG	-	8	1.5	4	single	-	No layer	Silicone oil	300 K, 500 K	Hardness
Ye [62]—Figure 11a	AA 7075	Nd:YAG	1064	5	5	1	-	75%	Al foil	BK7 glass	25 °C, 250 °C	Residual stress
Ye [63]—Figure 11b	AA 6061-T6	Nd-YAG	1064	5	1.5	2	4	75%	Al foil	BK7 glass	25 °C, 160 °C	Residual stress
Zhou [64]—Figure 12	Ti6Al4V	Nd-YAG	1064	10	12.73	3	-	50%	Al foil	Silicone oil	20–400 °C	Fatigue life
Ye [63]—Figure 13	AA 6061-T6	Nd-YAG	1064	5	1.5	2	4	75%	Al foil	BK7 glass	25 °C, 160 °C	Grain structure

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
