# Peer review of "The New Technologies Developed from Laser Shock Processing"

_materials, 2020, doi:10.3390/ma13061453_

Round 1

Reviewer 1 Report

The work of J. Wu et al is a review article on the development of methods and techniques based on laser shock processing. Fundamental questions of the interaction of pulsed laser radiation with matter were considered, and the theoretical model for the shock wave formation during plasma generation by laser radiation was presented. The methods of significant practical applications of laser materials processing, such as laser shock forming, nucleation, material hardening, marking and even imprinting in micro and nanoscale, are examined in sufficient detail. The work could be accepted to the  Materials journal with minor corrections.

1. Unfortunately, the calculation results according to the mathematical model described were not compared with experimental data. Since theoretical calculations are predictive, the results of comparison with the data of the corresponding experiments should be presented in this review article.

2. At the end of paragraph 3.2, the authors presented an excellent summary table with the main experimental parameters. An additional column with information on the achieved values ​​of the modified materials (hardness etc) parameters should be given in the table.

3. In the beginning of the 3.3 in the sentence “However, there exist some disadvantages in the LSM, ...” there is probably a misprint and the authors mentioned “LM” instead of “LSM”?

Author Response

The work of J. Wu et al is a review article on the development of methods and techniques based on laser shock processing. Fundamental questions of the interaction of pulsed laser radiation with matter were considered, and the theoretical model for the shock wave formation during plasma generation by laser radiation was presented. The methods of significant practical applications of laser materials processing, such as laser shock forming, nucleation, material hardening, marking and even imprinting in micro and nanoscale, are examined in sufficient detail. The work could be accepted to the  Materials journal with minor corrections.

  1. Unfortunately, the calculation results according to the mathematical model described were not compared with experimental data. Since theoretical calculations are predictive, the results of comparison with the data of the corresponding experiments should be presented in this review article.

Reply:This article mainly summary these new technologies developed from LSP. In section 2, we introduced the theoretical model of laser shock waves, which can give theoretical basis for researchers to study the effect of LSP on mechanical properties by numerical simulation method. In order to verify the accuracy of model, the researchers usually compare the mechanical properties of material (such as residual stress, hardness, etc) by numerical simulation method and experimental rather than comparison with the results for pressure of laser shock waves.

  1. At the end of paragraph 3.2, the authors presented an excellent summary table with the main experimental parameters. An additional column with information on the achieved values of the modified materials (hardness etc) parameters should be given in the table.

Reply: The experiment parameters is presented in the Table.1 According to the suggest of reviewer, we add an additional column to describe the modified materials parameters. We can see the detail revise as follow.

Tab.1 Main experiment parameters of LSP & WLSP provided by selected references reports in section 3.2

References

Material

Type of laser system

Laser parameters

Absorbing layer

Confining layer

Temperature

Materials parameters

Wavelength (nm)

Pulse width

(ns)

Power irradiance   (GW/cm2)

Diameter

(mm)

Repetition

Rate (Hz)

Overlap rate

Liao [60]-Fig.10(a)

AA6061

ND-YAG

1064

5

0.8-2.4

2

-

75%

Al foil

BK7 glass

20-160℃

Hardness

Tani [61]-Fig.10(b)

AISI 1042

Nd:YAG

-

8

1.5

4

single

-

No layer

Silicone oil

300K, 500K

Hardness

Ye [62]-Fig.11(a)

AA 7075

Nd:YAG

1064

5

5

1

-

75%

Al foil

BK7 glass

25℃, 250℃

Residual stress

Ye [63]-Fig.11(b)

AA 6061-T6

Nd-YAG

1064

5

1.5

2

4

75%

Al foil

BK7 glass

25℃, 160℃

Residual stress

Zhou [64]-Fig.12

Ti6Al4V

Nd-YAG

1064

10

12.73

3

-

50%

Al foil

Silicone oil

20-400℃

Fatigue life

Ye [63]-Fig.13

AA 6061-T6

Nd-YAG

1064

5

1.5

2

4

75%

Al foil

BK7 glass

25℃, 160℃

Grain structure

  1. In the beginning of the 3.3 in the sentence “However, there exist some disadvantages in the LSM, ...” there is probably a misprint and the authors mentioned “LM” instead of “LSM”?

Reply: Thanks for your suggest, the corrected word is “LM” instead of “LSM”. And the detail revise as follow.

However, there are exist some disadvantages in the LM

Reviewer 2 Report

I find this review paper interesting, it covers new applications of Laser Shock Processing (LSP).

My only concern is that I can not understand the low number of publications from different groups like A. Clauer; Y. Sano; R. Fabbro, L. Berthe, P. Peyre; J.L. Ocana, M. Morales; G. Gomez, C. Rubio; M. Schmidt.

I think that at least in the introduction there should be a better overview of the research groups that started the development of this technology.

I think that other applications are missing like LSP in Additive Manufacturing, LSP for Spall studies, ...

I think that a deeper study would be needed in order to have the quality to publish it. 

Author Response

 I find this review paper interesting, it covers new applications of Laser Shock Processing (LSP).

(1) My only concern is that I can not understand the low number of publications from different groups like A. Clauer; Y. Sano; R. Fabbro, L. Berthe, P. Peyre; J.L. Ocana, M. Morales; G. Gomez, C. Rubio; M. Schmidt.

Reply: In this work, we mainly review the new technologies developed from LSP. And the publications from different groups like A.Clauer et al.are mainly study the effect of LSP on the materials (traditional application) rather than the new technologies developed from LSP. So the number of these publications is relatively low. in addition, according the review of reviewer,  we will add some references from these groups. And the detail references can be seen as follow.

[18] Qiao H C, Gao Y, Zhao J B, et al. Research process of laser peening technology[J]. Chinese Journal of Nonferrous Metals, 2015, 25(7):1744-1755

[19] Lu G X, Liu H , Lin C H, et al. Improving the fretting performance of aero-engine tenon joint materials using surface strengthening[J]. Materials Science and Technology, 2019(2):1-8.

[20] Peyre P, Fabbro R, Berthe L, et al. Laser shock processing with small impacts[J]. Proceedings of Spie the International Society for Optical Engineering, 1996, 2789:125-132.

[30] Fairand B P, Clauer A H. Laser Generation of High-amplitude Stress Waves in Materials[J]. Journal of Applied Physics, 1979, 50(3):1497-1502.

[31] Fairand B P, Clauer A H. Effect of water and paint coatings on the magnitude of laser-generated shocks[J]. Optics Communications, 1976, 18(4):588-591.

[43] Braisted W, Brockman R. Finite element simulation of laser shock peening[J]. International Journal of Fatigue, 1999. 21: 719-724.

(2) I think that at least in the introduction there should be a better overview of the research groups that started the development of this technology.

Reply: According to the review from reviewer, the introduction add some summary, see as follow (the revise was marked in red).

For example, after the LSP treatment, the fatigue life of the vane-integrated disk can be improved about 4-6 times, which shows the great development prospect for LSP [18]. As a novel surface hardening treatment technology, LSP has the advantages of great hardening effect, high processing precision and good controllability of programming [19]. Compared with other normal surface hardening treatment technologies such as shot peening (SP), rolling and low plasticity burnishing, the depth of plastic deformation by LSP treatment can be reached to over 1mm, which is much higher than that by other methods with the depth of 75-250μm at most [18, 20]. In addition, LSP has the obvious advantages in maintaining the smooth surface morphology. For example, the 7075 Al alloy were treated by SP and LSP, the surface roughness of the SP-treated sample is 5.7μm, while that of the LSP-treated sample is 1.1μm [21].

(3) I think that other applications are missing like LSP in Additive Manufacturing, LSP for Spall studies, ...

Reply: In this work, the new technologies developed are mainly introduced from the aspect of LSF, WLSP, LSM and LSI. About the other applications such as LSP in Additive Manufacturing, LSP for Spall studies and so on, which are very interesting too, but we cannot find proper names to name them as a new technology developed from LSP. Thanks for your suggest, we will take more time to study other applications.

  • I think that a deeper study would be needed in order to have the quality to publish

Reply: we add deeper analysis in this manuscript, and the revises were marked in red in this revised manuscript.

In section 2.1:

At present, the process structure of constrain layer and absorbing protective layer is the most typical structure for the LSP treatment [29]. Whether the constrain layer and absorbing protective layer, its thickness should be suitable. The thickness of constrain layer can affect the transmittance of laser and the pressure of laser-induced shock waves, the lower thickness can increase the transmittance but easily cause the breakdown of the pressure, and the suitable thickness of constrain layer such as water for most of LSP process is about 1.2-2mm [10, 30]. And the absorbing protective layer can improve the absorption rate of laser energy and increase the peak pressure of laser-induced plasma shock waves as well as protect the materials from thermal melting. And the related research showed that there are exist a critical thickness of absorbing protective layer, the higher thickness can lead the pressure losses of laser-induced plasma shock waves, but the lower thickness can lead the thermal melting in the near-surface layer of material and the formation of rougher impact pits, and the common thickness of absorbing protective layer such as black tape is about 100μm [1, 31]. Therefor, in the LSP treatment, the suitable thickness of constrain layer and absorbing protective layer can improve the transmittance and pressure of laser-induced plasma, resulting in obtain the better hardening effect [32].

In section 2.2:

As observed from equation (5), we can know that the peak pressure of shock wave is proportional to the square root of laser power irradiance. In addition, the laser power irradiance obeys the quasi-Gaussian distribution over time, so the peak pressure of shock waves also obey the quasi-Gaussian distribution over time [24]. And the related reference [43] tell us that pulse width of the shock waves load is approximately 2-3 times that of laser pulse width.

In section 3:

In this section, we importantly introduce the new technologies developed from LSP from the aspect of laser shock forming, warm laser shock processing, laser shock marking and laser shock imprinting.

In section 3.2:

So the WLSP-treated samples will have a greater stability of dislocation structure. Due to the increase of surface hardness, compressive residual stress, depth of residual stress layer and the stability of dislocation structure, which will inhibit the generation and propagation of cracks, resulting in improve the stability of residual stress [68]. Compared with the LSP-treated samples, the compressive residual stress lever of the WLSP-treated sample is higher than that, and the effect of DA and DSA can provide the key contribution to the improvement [63, 69]. In conclusion, the material strength and stability of residual stress by WLSP treatment are higher than that by LSP treatment.

Reviewer 3 Report

Dear authors,

Please see attached review.

Respectfully,

Author Response

I reviewed the article “The new technologies developed from laser shock processing,” Jiajun Wu, Jibin Zhao, Hongchao Qiao, Xianliang Hu, Yuqi Yang, for consideration of publication in materials/MDPI.

The manuscript communicates applications of laser-induced shocks for materials processing including advanced surface treatment. There are but a few suggestions for edits, in random order:

  1. Figure 1: The shock wave is shown to enter the material only. However, there is also a shock wave emanating towards the laser side (back-reflection side). It is also important to specify ‘high’ and ‘short’ text and indicate the thickness of the layer in the figure. Important: nanosecond, femtosecond, and peak irradiance.

Reply: Figure 1 is the schematic of the principle of LSP. And the principle of LSP tell us that only the shock wave propagating inside the material can lead to plastic deformation in the near-surface of material, so the shock wave is shown to enter the material only in Figure 1. Due to Fig.1 is referred from ref[22], so about the specify “high” and “short” text and indicate the thickness of the layer will be discussed in section 2.1. The detail discussion can be seen as follow.

At present, the process structure of constrain layer and absorbing protective layer is the most typical structure for the LSP treatment. Whether the constrain layer and absorbing protective layer, its thickness should be suitable. The thickness of constrain layer can affect the transmittance of laser and the pressure of laser-induced shock waves, the lower thickness can increase the transmittance but easily cause the breakdown of the pressure, and the suitable thickness of constrain layer such as water for most of LSP process is about 1.2-2mm. And the absorbing protective layer can improve the absorption rate of laser energy and increase the peak pressure of laser-induced plasma shock waves as well as protect the materials from thermal melting. And the related research showed that there are exist a critical thickness of absorbing protective layer, the higher thickness can lead the pressure losses of laser-induced plasma shock waves, but the lower thickness can lead the thermal melting in the near-surface layer of material and the formation of rougher impact pits, and the common thickness of absorbing protective layer such as black tape is about 100μm. Therefor, in the LSP treatment, the suitable thickness of constrain layer and absorbing protective layer can improve the transmittance and pressure of laser-induced plasma, resulting in obtain the better hardening effect.

  1. Figure 2: Fig. (a) is confusing: How would the plasma affect the substrate?

Reply: The LSP is a novel surface hardening technology by utilizing the stress effect generated by laser plasma shock waves. In Figure 2 (a), the plasma can induce shock waves with the peak pressure of GPa order, When the laser-induced plasma shock wave is loaded on the surface of metallic material, uniaxial stress will be generated along the propagating direction of shock wave, which will lead to plastic deformation in the LSP area. In Figure 2 (b), after the laser-induced plasma shock wave is over, the plastic deformation region will be limited and restricted by the surrounding material, so a biaxial compressive residual stress field will be generated, which is paralleled to the LSP surface.

  1. Figure 3 and text describing laser-supported combustion and detonation waves: There is need for references, specifically: L.J. Radziemski, D.A. Cremers (Eds.), Laser-Induced Plasmas and Applications, Marcel Dekker, New York, 1989. In addition, please include Ref. Harilal, S.S.; Miloshevky, G.V.; Diwakar, K.; LaHaye, N.; Hassanein, A. Experimental and computational study of complex shockwave dynamics in laser ablation plumes in argon atmosphere. Phys. Plasmas 2012,19, 083504.

Reply: Thanks for your suggest, we add these advise references in this manuscript. And the detail reference can be seen as follow.

[35] L. J. Rakziemski, D.A. Cremers (Eds.), Laser-induced Plasmas and Applications, Marcel Dekker Inc, New York, 1989.

[36] Harilal, S.S.; Miloshevky, G.V.; Diwakar, K.; LaHaye, N.; Hassanein, A. Experimental and computational study of complex shockwave dynamics in laser ablation plumes in argon atmosphere. Phys. Plasmas, 2012, 19(8):083504.

  1. Near equations (7) and (8): Please include references to a standard optics textbook.

Reply: The references near equations (7) and (8) are showed as follow.

[39] Ma D Y. Theoretical foundation of modern acoustics [M]. Science Press, Beijing, 2004.

[40] Renk K F. Laser Principle[M]// Basics of Laser Physics. Springer Berlin Heidelberg, 2012.

  1. Figure 4: needs units! It is also important to clarify shockwave characteristics inside the medium, including use of propagation speeds, and discussion of phenomena that occur at various time scales, e.g., consider temperature rise and subsequent formation of pressure waves, and shockwave characteristics outside the medium, see comments in point 3) above.

Reply: we add the unit in Figure 4, can see detail in manuscript.

In addition, we add some discussion about the time law of shock waves. The detail revise as follow.

As observed from equation (5), we can know that the peak pressure of shock wave is proportional to the square root of laser power irradiance. In addition, the laser power irradiance obeys the quasi-Gaussian distribution over time, so the peak pressure of shock waves also obey the quasi-Gaussian distribution over time. And the related reference [] tell us that pulse width of the shock waves load is approximately 2-3 times that of laser pulse width.

  1. Below Equation 5: Please correct: Use ‘irradiance’ – laser power density would imply power per cc. Please also consider using ‘peak’ irradiance at other occasions, e.g., see text above Fig. 10.

Reply: The laser power density below equation 5 has been corrected to use the “irradiance”. And the corrected word was marked in red in the manuscript.

  1. As the laser interacts with the material, critical density (of the order of 10^21/cc electrons for 1064-nm radiation) is reached. This is also associated with a higher temperature than 10^4 K. It will help to include comments about critical density, and perhaps include comments regarding femtosecond material processing over and above nanosecond materials processing.

Reply: Thanks for your review or advise, we will consult the relevant literature and do further researches.

Round 2

Reviewer 2 Report

You have clarify the scope of the review. The review is about new technologies developed from LSP: laser shock forming (LSF), warm laser shock processing (WLSP), laser shock marking (LSM) and laser shock imprinting (LSI).

It is not a review about every new application based on LSP, and I understand that those have been chosen as more developed than some other.

In the introduction many references from the first groups have been added.

The explanations have been improved in every of the points asked in the first review.

I think that with all the changes done in the paper it can be published as it is..